



# Combined GNSS reflectometry/refractometry for automated and continuous in situ surface mass balance estimation on an Antarctic ice shelf

Ladina Steiner[1], Holger Schmithüsen[1], Jens Wickert[2,3], and Olaf Eisen[1]

[1]Alfred Wegener Institute, Helmholtz Centre for Polar and Marine Research, Am Alten Hafen 26, 27515 Bremerhaven, Germany
[2]GFZ German Research Centre for Geosciences, Telegrafenberg, 14473 Potsdam, Germany
[3]Technische Universität Berlin, 10553 Berlin, Germany

**Correspondence:** Ladina Steiner (ladina.steiner@alumni.ethz.ch)

**Abstract.** Reliable in situ surface mass balance (SMB) estimates in polar regions are scarce due to limited spatial and temporal data availability. This study aims at deriving automated and continuous specific SMB time series for fast moving parts of ice sheets and shelves (flow velocity $> 10\,\mathrm{m\,a^{-1}}$) by developing a combined Global Navigation Satellite Systems (GNSS) reflectometry and refractometry (GNSS-RR) method. In situ snow density, snow water equivalent (SWE), and snow deposition or erosion are estimated simultaneously as an average over an area of several square meters and independent on weather conditions. The combined GNSS-RR method is validated and investigated regarding its applicability on a moving, high latitude ice shelf. A combined GNSS-RR system was therefore installed in November 2021 on the Ekström ice shelf (flow velocity $\approx 150\,\mathrm{m\,a^{-1}}$) in Dronning Maud Land, Antarctica. Reflected and refracted GNSS observations from the site are post-processed to obtain snow accumulation (deposition and erosion), SWE, and snow density estimates with a 15 min temporal resolution. Results of the first 16 months of data show a high level of agreement with manual and automated reference observations from the same site. Snow accumulation is derived with an uncertainty of around 9 cm, SWE around $40\,\mathrm{kg\,m^{-2}\,a^{-1}}$, and density around $72\,\mathrm{kg\,m^{-3}}$.

This pilot study forms the base for extending observational networks with GNSS-RR capabilities, in particular in polar regions. Regional climate models, local snow modelling, and extensive remote sensing data products will profit from calibration and validation based on such in situ time series, especially if multiples of such sensors will be deployed over larger regional scales.

## 1 Introduction

The potential contribution of the Antarctic Ice Sheet to sea level rise is significant as it contains approximately 80 % of the world's freshwater (Vaughan et al., 2013; King et al., 2012). Ocean warming, variation in atmospheric circulation patterns, and enhanced atmospheric moisture lead to increased snowfall which dominates the observed positive trend in the East Antarctic Ice Sheet mass balance (Shepherd et al., 2012). Currently, the mass gain of the East Antarctic Ice Sheet potentially exceeds the increased mass loss of the West Antarctic Ice Sheet (e.g. Medley and Thomas, 2019), thus mitigating global sea level rise.



Davison et al. (2023) "emphasize the important impact of extreme snowfall variability on the short-term sea level contribution
from West Antarctica".

However, surface mass balance (SMB) estimates remain a significant uncertainty factor in ice sheet mass balance compu-
tation and projections due to the scarce spatial and temporal availability of in situ data (van den Broeke et al., 2017; van den
Broeke et al., 2009). Rising global temperatures are projected to lead to increased solid ice discharge, surface melt and SMB,
necessitating continuous time series of in situ snow density, snow water equivalent (SWE) and snow deposition or erosion
for correct SMB estimates and, thus, sea level rise predictions (e.g., Gardner et al., 2013; Hanna et al., 2008). Future mass

balances are therefore provided by prognosis for various scenarios using regional climate models (RCM), which are affected
by uncertainties from applied density assumptions (e.g., van Wessem et al., 2018). Validation and calibration are based on very
limited annual accumulation estimates from radio-echo sounding, firn and ice cores, snow pits, or unevenly distributed weather
stations with limited information about snow or firn conditions.

Extensive observations of SMB are a challenging task due to the heterogeneity of snow distribution caused by inhomoge-

neous snow deposition and ablation. Large-scale Antarctic ice sheet changes and snow observations can be derived from space-
borne radar interferometry (e.g., Rignot et al., 2011). Gravity observations (e.g., Soerensen and Forsberg, 2010; Velicogna and
Wahr, 2005) provide direct mass changes. Repeat laser altimetry, e.g. from airplanes or satellites, can quantify snow or firn
surface elevation changes (e.g., Markus et al., 2017; Helm et al., 2014). These remote sensing techniques cannot provide direct
estimates of SMB, snow or firn density, and SWE and thus necessitate accurate and reliable in situ data for calibration and

validation. Time series of surface density, averaged over local areas, are spatially representative on the polar plateau and of
utmost importance to link remote sensing derived elevation or volume changes to mass balance estimations  (e.g., Veldhuijsen
et al., 2023; Heilig et al., 2020; Weinhart et al., 2020). Continuous time series support increased understanding of temporal
changes in accumulation and melt, allowing improvements of RCM as well as polar snow modelling (e.g., Heilig et al., 2018).

Smith et al. (2017) provide a detailed summary and comparison of terrestrial, airborne, and spaceborne techniques.Lenaerts

et al. (2019) and Eisen et al. (2008) provide in-depth reviews of different in situ observation techniques used for SMB estima-
tion in polar regions. Manual in situ techniques such as snow pits are laborious and destructive, and have a low temporal and
spatial resolution. Moreover, they are affected by considerable uncertainties and suffer from irregular revisiting times in logis-
tically inaccessible regions, like the Antarctic or Greenland ice sheets. Ice core data analyses enable temporal quantification
of accumulation variabilities (e.g., Vandecrux et al., 2019), but for instance only allow to indirectly reconstruct the temporal

evolution of SWE. Automated in situ echo sounding, laser distance sensors, camera observing systems (e.g., Arslan et al.,
2017), terrestrial laser scanning (Prokop, 2008), radar interferometry (e.g., Frey et al., 2018; Leinss et al., 2015), or GNSS re-
flectometry (Larson and Small, 2016; Jin and Najibi, 2014; Larson et al., 2009b) deliver snow depth for seasonal snow cover on
a solid ground, but only provide accumulation on ice sheets, shelves and glaciers. Snow pillows or scales (Johnson et al., 2015;
Beaumont, 1965), cosmic ray sensors (Gugerli et al., 2019; Schattan et al., 2017) and acoustic sounding (Kinar and Pomeroy,

2015) provide SWE. Upward-looking radar systems yield snow depth, snow density, wetness, and SWE (Heilig et al., 2020;
Schmid et al., 2015). These existing observation techniques have either low spatial resolution being insensitive to snowpack
variability, or sensitive optical and moving parts limit their long-term polar application (Gutmann et al., 2012). Following the



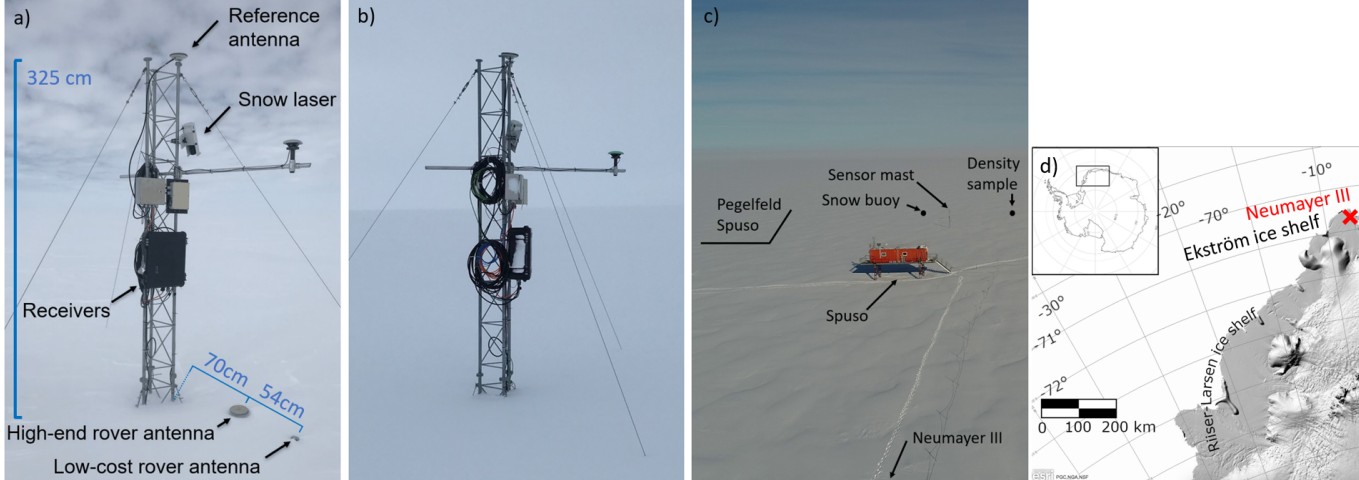

**Figure 1.** a) The GNSS-RR setup consists of one GNSS reference antenna and two GNSS rover antennas placed on the firn surface (21.11.2021). The rover antennas are physically connected to the sensor mast via a lateral boom (hidden below the snow surface). b) GNSS-RR setup with both rover antennas already covered by the accumulated snow (23.11.2021). c) Situation of the test site. d) Location of the test site at the Ekström ice shelf, Dronning Maud Land, Antarctica (modified from Jakobs et al., 2019).

idea of Limpach et al. (2013) and Gschwend (2012), refracted GNSS signals from antennas buried underneath a snowpack are recently investigated for SWE determination of seasonal snow on stable ground (Steiner et al., 2022; Capelli et al., 2022; Steiner et al., 2020, 2019; Koch et al., 2019; Henkel et al., 2018; Steiner et al., 2018b, a).

The present study investigates the potential of a combined GNSS reflectometry and refractometry method (GNSS-RR) for an accurate, automated, and continuous quantification of in situ SMB time series. The aim is to simultaneously estimate snow deposition or erosion (and thus net specific SMB), SWE, and snow density with a high temporal resolution and independent on weather conditions. The developed combined GNSS-RR method is evaluated on the fast moving Ekström ice shelf in Antarctica.

An overview about the combined GNSS-RR setup and available reference data is given in Sect. 2 while Sect. 3 summarizes the GNSS-RR method. Section 4 describes the results of the snow accumulation (Sect. 4.1), the SWE (Sect. 4.2), and the snow density estimation (Sect. 4.3) using the combined GNSS-RR method, followed by a discussion in Sect. 5. Finally, conclusions are drawn in Sect. 6.

# 2 Experimental setup and reference data

## 2.1 Experimental setup

A combined GNSS-RR system is designed and installed on the Ekström ice shelf, Dronning Maud Land, Antarctica, providing data for the first-time development and evaluation of the combined GNSS-RR method based on reflected and refracted GNSS




**Table 1.** Specifications of the deployed high-end and low-cost multi-frequency and multi-system GNSS sensors (LeicaGeosystems, a, b; Ublox; Emlid). The Raspberry Pi (RaspberryPi) is used for remote control and data transfer of the low-cost receiver.

| System | Sensor | Dimension | Weight | Average power consumption |
|--------|--------|-----------|--------|---------------------------|
| High-end | Leica AS10 antenna | 170 x 62 mm | 0.44 kg | - |
| High-end | Leica GR10 receiver | 220 x 200 x 94 mm | 1.67 kg | 3.6 W (150 mA @24 V) |
| Low-cost | u-blox ANN-MB1 antenna | 60 x 82 x 22.5 mm | 0.173 kg | - |
| Low-cost | Emlid Reach M2 receiver | 56.4 x 45.3 x 14.6 mm | 0.035 kg | 1 W (200 mA @5 V) |
| Low-cost | Raspberry Pi 4.B computer | 85 x 56 x 20.5 mm | 0.046 kg | 6 W (1200 mA @5 V) |

signals. Figure 1a shows the combined GNSS-RR system, which moves at ≈ 150 m a⁻¹ with the ice shelf towards NNE
(Fig. 1d). The setup is mounted on an already existing sensor mast from the Meteorological Observatory Neumayer of the
Alfred-Wegener-Institute Helmholtz Centre for Polar and Marine Research (AWI). The mast is located close to AWI's air
chemistry (Spuso) observatory (Fig. 1c), 1.5 km South of the German Antarctic research station Neumayer III (Wesche et al.,
2016).

The GNSS-RR setup consists of a GNSS reference antenna and a high-end and low-cost GNSS rover antenna placed on the
firn surface. The rover antennas are buried by snow accumulating on top of the antennas (Fig. 1b). It is mandatory that the
rover antennas are mechanically connected to the reference antenna when deployed on a moving ground surface. The physical
connection enables the correct separation of SWE induced effects and station height changes due to ground movements in the
GNSS parameter estimation as both parameters are highly collinear with 99,7 % (Steiner et al., 2020). The rover antennas are
physically (mechanically) fixed to the sensor mast via another lateral boom. Note that the data from a second GNSS reference
antenna, mounted on a lateral boom to the sensor mast (Fig. 1a) is not used in the present study.

The specifications of the deployed GNSS-RR system are summarized in Table 1. A Raspberry Pi computer is used for
remote control and data transfer of the low-cost receiver. The GNSS-RR system collects multi-frequency and multi-system
(GPS, GLONASS, Galileo) GNSS signals. The GNSS antenna mounted on top of the sensor mast serves as a GNSS reference
for differential processing and additionally tracks reflected GNSS signals (1 Hz sampling rate) from the surrounding snow
surfaces. The buried GNSS antennas collect refracted GNSS signals (30 s sampling interval) influenced by the accumulated
snow above the antennas from all elevation angles. Power and Ethernet supply are provided by AWI's nearby air chemistry
observatory (Spuso) which is connected to the Neumayer III station.

## 2.2 Reference data

Ground truth data for snow deposition and erosion (snow accumulation) are provided with a 1 min sampling interval by a
laser distance sensor (Jenoptik SHM30) from the same sensor mast (Schmithüsen, 2023). These data are resampled to 15 min
to match the temporal resolution of the GNSS derived results. Nearby (distance ≈ 20 m) snow buoy observations (Nicolaus
et al., 2021; Grosfeld et al., 2016), consisting of four sonic rangers (MaxBotix HRXL-MaxSonar-WR3) are furthermore used





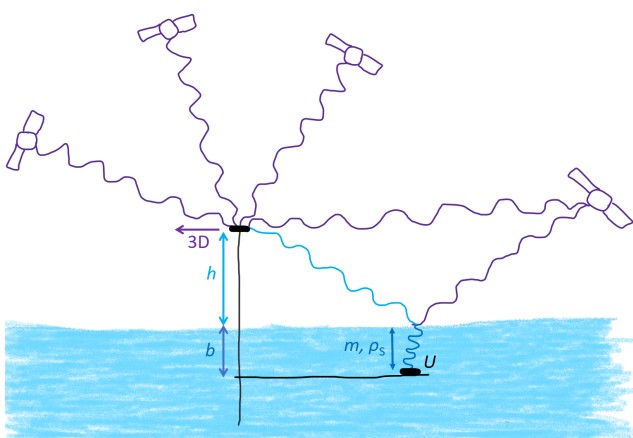

**Figure 2.** Schematic overview of the GNSS-RR measurement principle applied on a moving ground. Direct (purple), reflected (light blue), and refracted (dark blue) GNSS signals are collected to estimate snow accumulation ($b$), SWE ($m$), and the density of snow ($\rho_s$) above a buried GNSS rover antenna. $U$ is the rover Up coordinate and $h$ the height difference of the GNSS base antenna to the reflective surface.

for reference. Additional ground truth is provided weekly by 16 accumulation stakes from the close by "Pegelfeld Spuso" (distance $\approx 200$ m). Monthly manual snow accumulation, SWE, and snow density observations from the upper layer (first meter) are taken at the "density sample location" (distance $\approx 50$ m) at the same test site (Fig. 1c). A relative uncertainty of 10 % is added as an error bar in all plots to indicate the general accuracy of such manual observations. Continuous snow accumulation data from the laser distance sensor is additionally converted to SWE by linearly interpolating the monthly density observations to enable an accuracy estimate for the GNSS-RR derived SWE time series.

## 3 Combined GNSS-RR method

A combined GNSS-RR method using reflected and refracted GNSS signals (Figure 2) is applied and investigated regarding its potential for simultaneous, continuous, and accurate estimation of snow accumulation, SWE, and snow density.

### 3.1 GNSS reflectometry for snow accumulation estimation

In situ snow accumulation (deposition and erosion) is estimated using ground based GNSS reflectometry (e.g., Jin and Najibi, 2014; Larson et al., 2009b). Direct GNSS signals and GNSS signals reflected off a snow/firn surface are analyzed in order to measure the difference in height $h$ from the GNSS reference antenna to the reflective surface. The influence of the reflected signals on the signal strength depends on the path extension with respect to the direct signals. The overlay of the direct and the reflected GNSS signals creates an interference pattern of the signal strengths when satellites move across the sky. The frequency $f$ of these multipath oscillations are related to $h$ (with $\lambda$ being the GNSS wavelength of the observed satellite system; between $18 - 26$ cm):



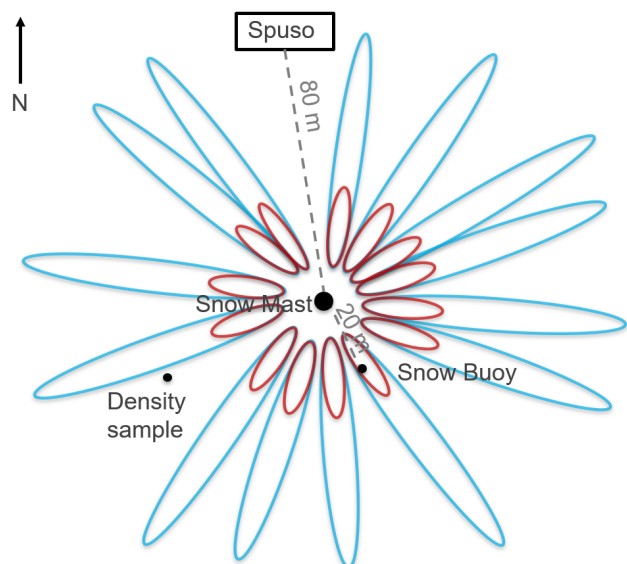

**Figure 3.** Fresnel reflection ellipses illustrating the area of received reflected GNSS signals around the reference antenna.

$$h = \frac{f\lambda}{2} \tag{1}$$

The snow accumulation $b$ is given by the change in height ($h$). The area around the GNSS reference antenna used to estimate snow accumulation is determined by the first Fresnel reflection ellipses as shown in Fig. 3 for the experimental setup. The size of these ellipses depends on the height of the reference antenna above ground and the analyzed incident angles.

The 1 Hz GNSS reference data is post processed using the open source gnssrefl package from Larson (2021). Best results are achieved by selecting GPS, GLONASS, and Galileo second frequencies within an elevation range of $5 - 30$ degrees. The reflector height was set to $2 - 8$ m to allow automated processing independent of regular sensor mast extensions of approximately 3 m. Distracting low elevation signals bending around or reflecting off the close by air chemistry observatory (Spuso) are excluded based on their azimuthal range. Outliers in the snow accumulation results are removed using a statistical threshold based on the standard deviation ($3\sigma$). The resulting time series is filtered by a moving median over 24 h. The derived results are resampled to 15 min values to enable the combination with the SWE estimates from GNSS refractometry.

## 3.2 GNSS refractometry for SWE estimation

The GNSS refractometry method based on the biased Up component (Steiner et al., 2022) is applied for in situ SWE estimation. Signals received by an antenna buried underneath the snowpack are delayed while propagating in a snow/firn layer. These delays bias the estimated antenna position, especially the vertical Up (height) component $U$. The resulting height deviation



from a phase-based differential GNSS position estimation provides information on the change in SWE (i.e., snow mass change $\delta m$), as both parameters are highly collinear with 99,7 % (Steiner et al., 2020):

$$\delta m \approx \delta U \tag{2}$$

The $\delta m$ above the buried rover antenna can thus be estimated based on the change in the estimated Up component $\delta U$ (Eq. (2)) of the rover coordinate. Equivalently, the Up component of the derived GNSS baseline between both antennas could be used directly. In contrary to previous GNSS refractometry studies, the present setup is situated on a fast moving ground. The correlation between the influence of snow on the GNSS signal delay and the ground movement is considered in the experimental setup (Sect. 2.1) by physically connecting the rover to the base antenna. If the physical height difference between both antennas would not be known and fixed, the SWE parameter could not be separated from the station height due to changes caused by the ground movement.

The GNSS refractometry data is post processed using the open source GNSS processing software RTKLIB version 2.4.3 b34 (Takasu, 2009). Although multi-system data is collected, the best results are achieved by only using multi-frequency GPS data for the SWE estimation. The additional use of Galileo and GLONASS observations increased the number of unusable solutions in case of our experiment. This is assumed to be caused by the applied processing software which was initially coded for GPS only and extended for multi GNSS systems afterward, potentially leading to problems in non-standard GNSS applications. Processing intervals of 15 min are applied for the differential GNSS processing with a very short baseline $(3-6\,\mathrm{m})$. All elevation observations are used to enhance the SWE estimation accuracy (Steiner et al., 2020). The reference coordinate is not fixed due to the ground movement, but selected automatically from each day's navigation position (available in the resulting RINEX observation file header) to guarantee an adequate initial coordinate for the differential processing. Similar to the snow accumulation estimation, outliers in the SWE results are removed using the $3\sigma$ statistical threshold. The 15 min SWE time series is filtered by a moving median over 24 h.

### 3.3 Combined GNSS-RR for snow density estimation

The in situ snow density $\rho_{\mathrm{s}}$ is derived by combining the individual results from GNSS reflectometry and refractometry:

$$\rho_{\mathrm{s}} = \frac{m \cdot \rho_{\mathrm{w}}}{b} \tag{3}$$

$\rho_{\mathrm{w}}$ is thereby the density of fresh water $(1000\,\mathrm{kg\,m^{-3}})$. Resulting daily densities lower than the density of new snow $(50\,\mathrm{kg\,m^{-3}})$ or higher than the density of firn $(830\,\mathrm{kg\,m^{-3}})$ are removed for plausibility. The SWE and snow density are determined as an average over an area of a few square meters around the antenna, depending on the height of snow above the buried rover antennas, the snow wetness, and the signal incidence angles.



## 4  Results

Combined GNSS-RR estimation results for snow accumulation (Sect. 4.1), SWE (Sect. 4.2), and snow density (Sect. 4.3) are
presented from the fast moving Ekström ice shelf in Antarctica. Results are compared to ground truth data from the same test
site (Sect. 2.1).

### 4.1  Snow accumulation

Figure 4a illustrates results from GNSS reflectometry using the open source gnssrefl package for the end of November 2021 to
April 2023. All snow accumulation estimates (transparent steel blue area) are overlaid by the median filtered snow accumulation
estimates (steel blue line). The snow accumulation varies significantly (standard deviation of 23 cm) over the reflection area
which covers a radius of up to 80 m around the reference antenna (Fig. 3). The variation is predominantly caused by strong
winds leading to spacial and temporal heterogeneity in snow deposition and erosion within the test site. The estimated snow
accumulation is directly compared to reference data from the laser distance sensor (dark blue) from the same mast. The GNSS
derived accumulation shows a very high level of agreement to the laser observations for the first weeks in Nov/Dec 2021 with
differences around 2 cm (Fig. 4b). Heavy storms in mid-December 2021 led to a strong increase of snow accumulation (up to
50 cm) as observed by the laser sensor. In contrary, the increase in accumulation derived by GNSS reflectometry is significantly
less with 32 cm for the same time period. This difference is on the order of the horizontal homogeneity of the snow surface and
may be explained by the different foot-prints of the two methods. This rationale is supported by comparable extreme values of
the GNSS estimate around this time. Note that the laser distance sensor broke down in the beginning of October 2022 and was
replaced in the end of December 2023, leading to a lack of data for this time period.

Additional reference data (Sect. 2.2) is available for comparison (Fig. 4c) from the nearby manual observations (dark blue
dots), snow buoy (gray curves), and 16 stake observations (colored dotted lines). Strong variation in snow accumulation is
observed between all reference sensors data with similar magnitudes as the difference in accumulation between the GNSS
derived results and the laser observations.

The accumulation observations decrease significantly after the mid-December event and last until February 2022. The GNSS
derived accumulation follows closely the trend observed by the laser distance reference sensor for the rest of the observation
period. The differences between the GNSS results and the laser vary around 35 cm. Overall results have a root mean square
error (RMSE) of 8.5 cm (Table 2) compared to the laser observations.

The accumulation results are compared to the laser observations in Fig. 4d based on a linear regression. The snow accu-
mulation estimated by GNSS reflectometry is highly correlated to the laser observations, with the Pearson cross-correlation
coefficients (r) of 0.98. The offsets induced by the snow deposition and erosion heterogeneity are visible in the linear fit. Table 2
lists the regression coefficients.



**Figure 4.** a) Time series of median GNSS reflectometry derived snow accumulation (steel blue line) overlaid by all accumulation estimates (steel blue area). b) Differences of all reference sensor observations to the median GNSS reflectometry estimates and c) visual comparison of all available reference data for November 2021 to April 2023. A relative uncertainty of 10 % is added as an error bar to the manual observations. d) Correlation of GNSS with the laser sensor accumulation.

**Figure 5.** a) Time series of median of GNSS refractometry derived SWE for the high-end and low-cost system, compared to reference data for November 2021 to April 2023. A relative uncertainty of 10 % is added as an error bar to the manual observations. b) Differences of the low-cost GNSS refractometry estimates and the manual and laser reference sensor observations to the high-end GNSS refractometry estimates. Correlation with c) manual observations and d) laser distance sensor data.



**Table 2.** Regression coefficients for the comparison of the snow accumulation estimated by GNSS reflectometry (GNSS-R) and observed by
the laser for 2021 – 2023. The linear regression fit is defined by the offset (o) and the slope (m). The number of samples are given by n.

| Reference data | System | o | m | r | n | RMSE |
|---|---|---|---|---|---|---|
| | | (cm) | | | | (cm) |
| Laser | GNSS-R | -10 | 1.12 | 0.98 | 952 | 8.5 |

## 4.2 Snow water equivalent

The GNSS refractometry derived SWE is median filtered and shown from the high-end (black) and low-cost (orange) system
in Fig. 5a for the end of November 2021 to April 2023. The median SWE time series is overlaid by the noise (standard
deviation per day; transparent black and orange) of the 30 s SWE estimation before filtering (3 kg m$^{-2}$ a$^{-1}$ for the high-end
and 6 kg m$^{-2}$ a$^{-1}$ for the low-cost system). The standard deviation of the median filtered SWE results from the high-end and
low-cost system is 2 kg m$^{-2}$ a$^{-1}$. Both GNSS derived SWE estimates strongly agree with each other until August 2022 with
noisier results from the low-cost system. Due to unknown reasons, differences between the high-end and low-cost system
increase afterward where the low-cost results start to underestimate the SWE up to 64 kg m$^{-2}$ a$^{-1}$ (Fig. 5b).

The derived SWE is directly compared to close by manual reference observations (blue dots). A relative uncertainty of
10 % is added as an error bar to the manual observations. Additional reference with a higher temporal resolution is provided
by the laser distance observations (dashed blue line). The observed snow accumulation data is thereby converted to SWE by
linearly interpolating available manual density measurements (Sect. 2.2). Note that the laser distance sensor broke down in the
beginning of October 2022 and was replaced in the end of December 2022, leading to a lack of data for this time period.

The GNSS refractometry derived SWE shows a very high level of agreement to the laser reference observations for the
first weeks of diminishing snow in Nov/Dec 2021 with differences around +/- 5 kg m$^{-2}$ a$^{-1}$ (Fig. 5b). As already visible in the
snow accumulation results (Sect. 4.1), heavy snow falls increased the SWE up to 200 kg m$^{-2}$ a$^{-1}$ (observed by the laser sensor)
or even 230 kg m$^{-2}$ a$^{-1}$ (observed manually) in mid-December 2021. The increase in SWE derived by GNSS refractometry
is significantly less with 105 kg m$^{-2}$ a$^{-1}$ for the same time period. The reason is similar to the difference observed for the
snow accumulation results and caused by the heterogeneity of the snow deposition and erosion within the test area. The GNSS
refractometry derived SWE follows closely the trend of the laser observations after mid-December. Differences to the laser
data vary within 193 kg m$^{-2}$ a$^{-1}$. Higher deviations (up to 204 kg m$^{-2}$ a$^{-1}$) are present compared to the manual observations.
Results from the high-end system have a RMSE of 32 kg m$^{-2}$ a$^{-1}$ to the manual and 43 kg m$^{-2}$ a$^{-1}$ to the laser observations
(Table 3). Results from the low-cost system have a RMSE of 36 kg m$^{-2}$ a$^{-1}$ to the manual and 38 kg m$^{-2}$ a$^{-1}$ to the laser
observations.

The SWE results are compared to the reference observations in Figure 4c,d using a linear regression. The SWE estimates
derived by GNSS refractometry are highly correlated to the manual and laser observations with r between 0.94 and 0.97





**Table 3.** Comparison of the SWE estimated by GNSS refractometry and observed by each reference sensor for 2021 – 2023. The linear regression is defined by the offset (o) and the slope (m); The number of samples is given by n.

| Reference data | System | o | m | r | n | RMSE |
| --- | --- | --- | --- | --- | --- | --- |
| | | $(\mathrm{kg\,m^{-2}\,a^{-1}})$ | | | | $(\mathrm{kg\,m^{-2}\,a^{-1}})$ |
| Manual | High-end | -103 | 0.84 | 0.97 | 15 | 32 |
| | Low-cost | -61 | 0.7 | 0.94 | 15 | 36 |
| Laser | High-end | -85 | 0.92 | 0.96 | 33'321 | 43 |
| | Low-cost | -59 | 0.82 | 0.96 | 33'321 | 38 |

(Table 3). The offsets induced by the snow deposition and erosion heterogeneity are visible in the linear fit. Table 3 lists the
regression coefficients.

### 4.3 Snow density

Combining the median-filtered, GNSS derived results for snow accumulation and SWE leads to estimates of snow density (temporal resolution of 15 min) for the observed area. Figure 6a illustrates the SWE and accumulation results from GNSS reflectometry and refractometry, respectively. Snow accumulation decreases in November and December 2021 followed by
snow deposition in mid-December 2021. Generally, snow deposition and erosion is dominating the change in SWE. It seems, however, that the snow settles during August and September 2022 after an increase in accumulation as the deposited snow decreases while the SWE stays nearly constant.

     The combined GNSS-RR derived snow density is shown from the high-end and low-cost system in Fig. 6b for the end of November 2021 to April 2023. Manual density observations and their accuracy are shown for reference. The GNSS-RR
derived densities are calibrated to the manual observation of 24 July 2022. Approximately one meter of snow accumulated on top of the buried GNSS antennas on that day which enables a correct match with the manual density observation of the upper one-meter layer (top 1 m) for better comparison. Results from the high-end and low-cost GNSS-RR method agree well with each other until August 2022. The low-cost system yields lower densities after August 2022 compared to the high-end results. Strong variations in the derived density time series are present until March 2022 for both systems compared to the
manual reference observations. During this time interval, the snow accumulation above the buried antennas was quite shallow (below 20 cm) and consequently the estimated SWE values very low. As the GNSS derived SWE estimation uncertainties are independent of the magnitude of the SWE itself, large density variations can result for small SWE values. Once the snow above the antennas is thicker, the combined GNSS-RR method enables feasible density estimates which agree well to the manual reference observations. The GNSS-RR derived estimates yields higher densities after October 2022 compared to the
manual observations with an increasing trend (Fig. 6c). The accumulation on top of the buried GNSS antennas exceeds one meter after October 2022. The larger amount of snow above the buried antennas ($b > 1$ m) could lead to the illustrated increase in the density observations compared to the manual observation of the top 1 m of the snowpack.



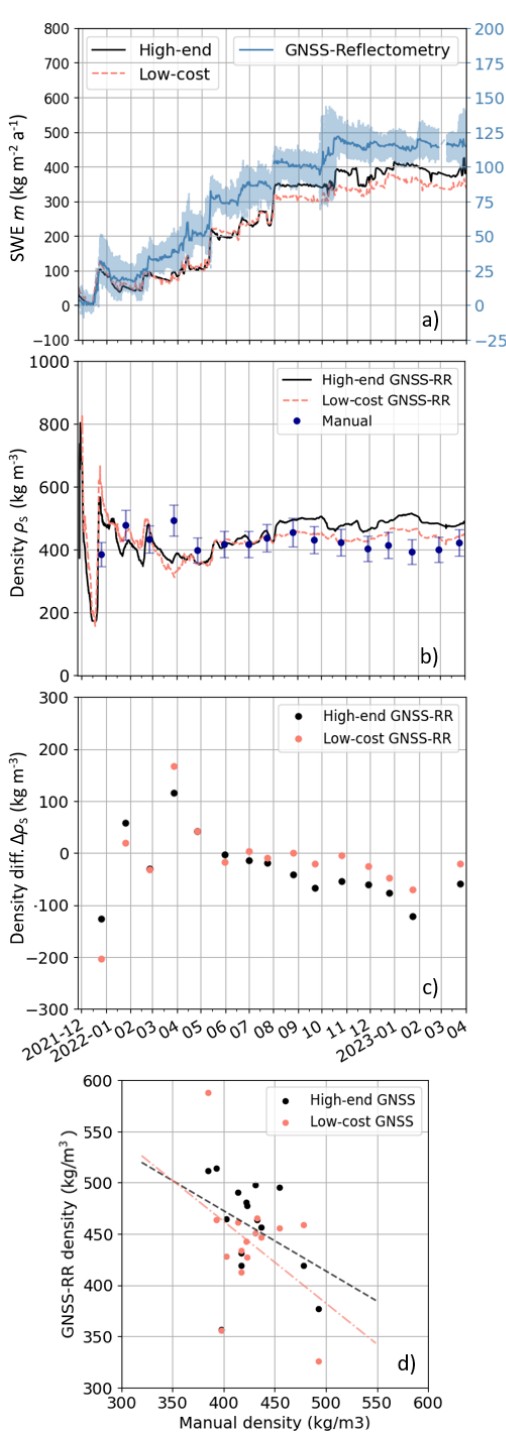

**Figure 6.** a) Comparison of GNSS reflectometry and refractometry derived accumulation and SWE time series. b) GNSS-RR derived density time series from the high-end and low-cost system compared to manual observations. c) Differences to and d) correlation with manual observations.





**Table 4.** Comparison of the GNSS-RR derived density and the manual observations for 2021 – 2023. The linear regression is defined by the offset (o) and the slope (m); The number of samples is given by n.

| Reference data | System | o | m | r | n | RMSE |
| --- | --- | --- | --- | --- | --- | --- |
| | | (kg m$^{-3}$) | | | | (kg m$^{-3}$) |
| Manual | High-end | 708 | -0.6 | -0.37 | 15 | 70 |
| | Low-cost | 782 | -0.8 | -0.42 | 15 | 74 |

The high-end and low-cost density results are compared to the manual reference observations in Figure 6d using a linear regression. The GNSS-RR derived density estimates show a low negative correlation to the manual observations, with r being -0.37 for the high-end and -0.42 for the low-cost system (Table 4). These results are highly influenced by the strong variation in the first months due to the shallow snow pack above the buried GNSS antennas. Table 4 lists the regression coefficients.

## 5  Discussion

Generally, the GNSS-RR derived snow accumulation, SWE, and density estimates agree well to the reference sensors data over the evaluated period of 16 months. All reference sensors measure on a point-wise, local scale and are situated at different locations within the experimental site (Sect. 2.2) with distances up to 200 m (stakes) from the GNSS-RR installation. Although the laser reference sensor is deployed on the same sensor mast as the GNSS-RR system, the measurement areas (spatial footprints) differ significantly. The laser observation is a point measurement whereas the spatial footprint of GNSS refractometry depends on the depth and permittivity of the accumulated snowpack above the buried GNSS antenna and the observed GNSS incidence angles. In the present case of using all GNSS observations from all available incidence angles, an area up to 2 m in diameter is observed for a 1 m dry snowpack above the buried GNSS antenna. GNSS reflectometry collects reflections even over a much larger area with a diameter up to 80 m around the GNSS base antenna for the analyzed incidence angles of 5 − 30 degrees and antenna heights up to 3 m above ground. Strong variation in snow accumulation caused by the heterogeneity in snow deposition and erosion over the observation area is therefore visible in all sensor observations due to the different sensor locations and spatial footprints. Higher deviations between the GNSS derived accumulation and SWE estimates compared to the laser observations are present after the strong accumulation event in mid-December 2021. These accumulation and SWE differences are of similar magnitudes as the deviations between all reference sensors observations and assumed to be caused by the heterogeneity of the snow deposition and erosion within the test area.

A main advantage of the combined GNSS-RR method is the possibility to continuously estimate the snow or firn accumulation, SWE, and density automatically at low-cost, allowing to support the understanding of surface related processes, such as firn densification processes, in future studies. Combined GNSS-RR can therefore support the identification and interpretation of the dominating surface related snow process, especially in case of decreasing snow surface levels. It is not possible to distinguish between snow erosion, compaction, and sublimation processes with pure accumulation observations, such as laser, buoy,



or stake observations. This interpretation can be supported thanks to the simultaneous observation of accumulation, SWE, and density with a high temporal resolution. Decreasing snow accumulation observed together with decreasing SWE can indicate

snow erosion or melting (e.g., end of December 2021 and January 2022, Figure 6a). Decreasing snow accumulation together with a stable SWE could be interpreted as snow settling, dominated by compaction and sublimation processes (e.g., August and September 2022, Figure 6a and b).

The GNSS-RR system is a promising method as the system is of small size and cost, easy to deploy, needs little maintenance, is passive and nondestructive, and enables automatic observations independent of weather conditions. Beside heightening the

sensor mast to compensate for yearly accumulating snow, no access is required during the measurement period which simplifies the application in remote polar environments. Power supply can thereby be a limiting factor. Point wise observations are additionally limited by their representativeness for the spatial variability and thus the understanding of SMB related processes on a larger scale. Further studies need to investigate the combined GNSS-RR method in more detail based on multiple dispersed stations at different experimental sites.

A minimum amount (20 cm) of snow above the buried GNSS antenna is shown to be a prerequisite to achieve reliable results for the density estimation. Previous studies show the feasibility of GNSS refractometry for accurate SWE estimation even for a very shallow snowpack above the buried antenna (Steiner et al., 2018a). Similar findings were obtained for GNSS reflectometry (e.g., Larson et al., 2009a). It is assumed here that the strong deviations in density estimation in the first months are due to the high heterogeneity within the test site. In combination with a very shallow snowpack, this leads to high relative uncertainties of

the individual snow accumulation and SWE estimates. Consequently, the uncertainties in the density estimation derived from a combination of these input values are propagated, leading to less reliable density estimates. This could be avoided by installing the GNSS rover antennas approximately 20 cm below the surface in the beginning. In this case, the snow accumulation, SWE, and the density above the buried antenna need to be measured to be able to calibrate the initial observation.

Another limitation is the inclination of the application surface as the height component of the GNSS baseline between the

base and rover antennas must be fixed. The system needs to be deployed in a very stable setup. Otherwise, e.g., if the system would be mounted on a tripod on a glacier surface with high surface melt, the setup could get tilted which changes the height difference between the base and rover. This change cannot be separated from influences due to snow above the buried GNSS antenna.

## 6   Conclusions

The present study illustrates the potential of a combined GNSS-RR method for accurate, simultaneous, and continuous estimation of in situ snow accumulation, SWE, and snow density time series. The combined GNSS-RR method was successfully applied on a fast moving, polar ice shelf. Snow accumulation results could be accurately determined using reflected GNSS observations. SWE was successfully estimated with a high temporal resolution (15 min) using GNSS refractometry based on the biased Up component. A high level of agreement to available reference data was achieved for both individual methods.

Combining results from both methods illustrated the potential of using a combined GNSS-RR approach for deriving in situ

snow densities with a 15 min resolution. A minimum amount (20 cm) of snow above the buried GNSS antenna was thereby shown to be a prerequisite to achieve reliable results.

The combined GNSS-RR approach could be highly advantageous for a continuous quantification of ice sheet and glacier SMB. The accumulation change of snow and firn can be derived using one single method by simultaneously estimating the
snow accumulation, SWE, and density with a high temporal resolution. Snow-hydrological modelling predicts local snow distribution, snow drift and melt, whereas climate models are used to predict mass balance changes on a regional scale. Both type of models could profit from such derived field measurements. The combined approach could also supplement or replace manual in situ data collection, leading to reduced expenses and enhanced temporal and spatial resolutions of the retrieved snow characteristics. Beside the application on a polar ice sheet, ice streams and shelves, simultaneous retrieval of these snow
characteristics in seasonal snow additionally supports public institutions, private companies, and environmental offices by providing fundamental data for managing the drinking and irrigation water supply, hydro-power energy supply, or assessing flood and avalanche risks. Point wise observations are, however, limited by their representativeness for the spatial variability and thus the understanding of SMB related processes on a larger scale.

Future research could further investigate the potential of linking such derived in situ density time series, available with a high
temporal resolution to extensive surface elevation observations (e.g., laser or radar altimetry) for improved SMB estimation on a larger scale. This could allow more reliable assessments and enhanced understanding of the contribution of SMB related processes driving future sea level rise.

*Code and data availability.*   Python code for preprocessing, processing, analyzing, and visualizing the GNSS-RR data is provided on Github (Steiner, 2023). Collected and analyzed multi-frequency and multi-system GNSS data are made publicly available at PANGAEA (Steiner
et al., 2023).

*Author contributions.*   Conceptualization, L.S., J.W. and O.E.; fieldwork, L.S. and O.E.; methodology, L.S. and J.W.; analysis and investigation, L.S.; reference data and accessibility to the Meteorological Observatory Neumayer, H.S.; draft preparation, L.S.; visualization, L.S.; glaciological know-how, expedition expertise, and scientific coordination, O.E.

*Competing interests.*   The authors declare no competing interest.

*Acknowledgements.*   L.S. was granted a postdoc.mobility fellowship (P400P2_199328/1) from the Swiss National Science Foundation. The research was also supported by the Helmholtz POF IV initiative. This project was strongly supported by many collaborators who all together enabled a successful outcome thanks to their valuable support and inputs.
The authors would like to thank Andreas Frenzel, engineer at AWI, for the valuable technical support in the experimental setup design stage.





We would also like to thank Markus Ramatschi (GFZ) for the expertise and preliminary work related to a pre-existing GNSS-R system at
the same sensor mast which simplified our system setup. We also thank Bernhard Richter, Vice President Geomatics at Leica Geosystems
AG, for supporting this project by providing the high-end GNSS equipment and support as part of the Hexagon's sustainability strategy.
A big thank you to the logistics experts Stefanie Klüver (AWI) and Isabelle Rümmele (Leica Geosystems AG) for the valuable support
with customs regulations regarding the shipping of the high-end GNSS equipment from Switzerland to the EU. Huge thanks to Martin Petri
and Norbert Anselm, observation to archive (O2A; Koppe et al., 2015) at AWI, for providing profound IT support, server infrastructure at
Neumayer III, data transfer to Germany, and the framework for the project dashboard for online data checking. Thank you to Peter Köhler,
senior scientist at AWI, for the strong help during the GNSS-RR system set up in Antarctica. Special thanks to the overwintering team
at Neumayer III (Linda Ort, Theresa Thoma, Paul Ockenfuss, Hannes Keck, and Nellie Wullenweber) for the on-site support, fieldwork,
and reference data observation. Thanks to Rolf Weller for the accessibility to the air chemistry observatory at Neumayer III. Last but not
least, a big thank you to the expedition organizers and all team members for the selection, contribution, and support of this project as part
of the German Antarctic expedition in 2021/22. Autonomous snow buoy measurements from Nov 2021 to April 2023 were obtained from
https://www.meereisportal.de (grant: REKLIM-2013-04).





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
