# Peer review of "Combined GNSS reflectometry/refractometry for automated and continuous in situ surface mass balance estimation on an Antarctic ice shelf"

_The Cryosphere, 2023_

## Author Comment (AC1)

***Specific Comments:***

1.  *The main challenge the authors faced in comparing different measurement series was the difference in the footprint of the individual measurements and, therefore, the impact of surface roughness on the results. As a reader I would like to see this issue discussed further, ideally supported by some data. The authors could add a more rigorous analysis of uncertainties. At present it is more or less covered by two sentences (section 5. line 252-257).*

    The paragraph is thoroughly extended by adding all observation accuracies from literature and further interpretation.

2.  *I would also like to know if outliers were filtered from the laser data? Lasers are subject to errors caused by blowing snow at the snow surface.*

    The laser observations are filtered by a moving median over 24 h to remove outliers and resampled to 15 min to match the temporal resolution of the GNSS derived. A sentence is added in section 2.2 (reference data).

3.  *Why do the high-end and low cost receiver time series diverge after August 2022? I think a hypothesis is needed (section 4.2, line 194).*

    Indeed, a hypothesis is missing in the text. We checked all possible error sources which are related to the GNSS refractometry processing, such as the antenna height, base coordinates, antenna calibration parameters, reference ellipsoid (WGS84), satellite availability and signal strengths. As GNSS refractometry is a relative observation method between the base and rover antenna, the processing is consistent for all data epochs and receivers, and there was no change in satellite signatures, we could exclude all such error sources. Since the offset is sudden and affects the receiver height, we assume a change in the effective physical height of the low-cost antenna. The antenna is screwed on the submerged lateral boom via a small vertical balise bar. Due to the overlying pressure of the snowpack and the cold, it could be possible that the screw has become loose and the antenna sank a few centimeters into the underlying snow. This can only be verified once the buried antennas are dug out in future (scheduled for season 2023/24). This has not been done to this stage in order to not disturb the snowpack.

    The text is adapted to include this hypothesis.

4.  *I note that according to figure 6 the low-cost data better tracks the manual density measurements after Aug. 2022 (section 4.3, line 228). This is not mentioned in the text and this omission should, in my option, be addressed.*

    A sentence is now added in section 4.3 to address the fit of the low-cost density results after August 2022. The order of the paragraph is rearranged for more clarification and context.

5.  *I would also expect the high-end receiver to better correlated with the laser data as the former seems directly under the latter while the low-cost antenna is further from the laser footprint.*

The pictures seem to mislead the interpretation of the position of the buried antennas to the laser footprint. The laser is oriented approximately towards 90 degrees (East) compared to the buried antennas and thus the high-end sensor is not directly below the laser.

6. *Section 4.1: I would like to know if there are relationships between GNSS satellite zenith and aspect, and the errors. Is there are directional component, related to, for example, the effect of the mast and prevailing wind on surface roughness?*

As mentioned in sections 3.1, azimuths where the GNSS-IR signals were bended around or reflected off the nearby air chemistry observatory were excluded from the GNSS-IR processing. Single azimuthal bins were not analyzed as it is intended to average all azimuthal observations to a) get the mean reference point around the mast to enable a comparison to the laser reference data, and b) filter all heterogeneities in surface roughness over the whole GNSS-IR footprint.

Single elevation bins cannot be formed as the GNSS-IR method uses all satellite tracks from 5-30 degrees elevation to be able to calculate the multipath oscillation frequency, needed for accumulation estimation.

7. *Should tiltmeters and power monitoring be added to the setup?*

If the system setup is prone to tilting and the relative height difference between the reference and rover antennas cannot be ensured, a tiltmeter would make sense to enable to monitor the physical baseline height component. Power monitoring could be added for remote self-sufficient locations.
The discussion section is now extended to include this suggestion.

8. *Would a comparison of your results and accuracies from different measurements (manual density, SWE; surface height/accumulation from lasers and sonic rangers) be appropriate using the literature? Perhaps as a short paragraph at the end of the discussion to provide additional context.*

Please see point 1. We have extended the first paragraph in the discussion section to include different measurement accuracies from literature to enable further discussion of the influence of the different footprints on the measurement uncertainties coupled to the heterogeneity of the snow pack around the antenna. We think that in that way your comment is addressed sufficiently.

*Technical Corrections:*

1. *In several places, especially in sections 2 and 3 you use the term "ground" to describe the ice shelf surface. Please use the term "surface" as it is not strictly speaking ground.*
The term "ground" is replaced by the term "surface" for all relevant parts connected to the ice shelf surface.

2.  *In section 2.1 it would be useful to know which reference ellipsoid was used: was it the same for all instruments? If not could there be an effect on data quality?*

    The same reference ellipsoid (WGS84) is used for all instruments. Thus, all results are consistent. This is now addressed in the manuscript by the extended text reffered to in comment 3.

3.  *p4. l80: could the arm for the buried GNSS antennas bend? Was it rigid?*

    The buried lateral boom is rigid and initially placed on very dense, wind-packed snow. Bending of the arm is thus assumed to be neglibible. A sentence is added in this section for clarification.

4.  *p5. l10; eq. 1: should h not be δh if it is a difference or change?*

    Eq. 1 refers to the relationship of the height (h) above the surface to the frequency of the multipath oscillations detected by GNSS reflectometry. The difference of this height (Δh) to the initial height over time then gives the change in accumulation. The respective sentence is adapted for clarification.

5.  *p.6 fig. 3. For the sake of completeness please explain the difference between cyan and mauve ellipses or what they represent (which antennas they represent). How were they calculated?*

    The figure caption is extended to clarify the meaning of the colors. The colors represent the Fresnel reflections ellipses for the limits of the used elevation range for GNSS reflectometry: Red (30°) and blue (5°). They are calculated by the first Fresnel reflection ellipsoid.

6.  *p7. eq. 3. Please remind the reader that m is SWE and b is accumulation. I have a short memory.*

    The reader is now reminded about the meaning of m and b in the respective section.

7.  *p9. fig. 4. (also figures 5 and 6): would it not be better to have sub-figures a) and b) across the top and c) and d) along the bottom of these figures? This might be pedantry, I know.*

    We chose the order of the subfigures carefully with the intention to enable the reader to easily compare and interpret the differences between the individual measurement methods with the absolute values based on the same x-axes.

---

## Author Comment (AC2)

1. *What is the reference antenna? It might be the same Leica AS10 as the high-end rover, but it should be mentioned. Same antennas usually give results matching better, thus some of the difference between high-end and low-cost rovers might stem from this. Also the antenna calibration might be a thing to check, if you want to improve the results, at least no calibration tables were mentioned in the text.*

   The reference antenna/receiver is similar to the rover antenna/receiver and oriented in the same direction to mitigate differences due to system setup and antenna phase center offset and variations. Antenna calibration files are also added in the processing to reduce these effects for the differential processing of the low-cost system. Thank you for pointing out that this information was missing in the text, which is now added in section 2.1.

2. *How is the mast attached to the moving ice, i.e. how is the stability of the monument ensured? I was also wondering, along with a colleague earlier, how you can ensure that the mast or the lever is not deforming inside the ice?*

   The triangular mast is considered highly stable as it is more than 15m deep in the firn/ice and tensed into the firn/ice with 3 steal ropes every 3m in height. Mast deformations or sinking within the shelf ice/firn are not relevant for the GNSS-RR method as it is a relative observation method between the reference and rover antennas. Deformation in the upper few meters is prevented by the three guy ropes every 3m. The buried lateral boom is rigid and initially placed on very dense, wind-packed snow. Bending of the arm is thus assumed to be negligible. A sentence is added in this section for clarification.

3. *One thing that could be added would be the number of satellites observed by the GNSS antennas. How many satellites are left when the elevation angle is limited for the reflectometry? And how many satellites there are generally available? This is one factor affecting the quality of the results as well and could maybe explain if there were problems with the data.*

   In general, there are 7-13 GPS satellites available and used for the refractometry processing. A sentence is added in the processing description in chapter 3. As the overall variation in satellite visibility is small, we chose to not include an extra figure with the number of available satellites.

   The number of satellites does not decrease for reflectometry as the experimental site is totally flat and all observations from the low elevation range are used, including the ascending and descending tracks.

4. *Some terminology is used abundantly, for example, line 129 "vertical Up (height) component U". For a geodesist, one would be enough, up component, height, or vertical component.*

   As a geodesist I agree. Since the paper is interdisciplinary and also focused on non-geodesists, we think that it is helpful to be explicit on the terminology for the first occurrence of "U" to reduce misunderstandings. See also the minor comment 6 of reviewer 1 ("short memory").

5. *Also the symbol* m *seems to mean both mass (in equations) and the slope (in tables).*

Indeed, the symbol "m" is used abundantly. The symbol for the slope is now changed to "a".

*I agree with colleague on the points regarding 1) the order of figures 4 and 5,* a *and* b *on top row and* c *and* d *below is much easier to read, and 2) some explanation why the results seem to lie nicely on top of each other before August 2022 and diverge after that.*

1) We chose the order of the subfigures carefully with the intention to enable the reader to easily compare and interpret the differences between the individual measurement methods with the absolute values based on the same x-axes.

2) We checked all possible error sources which are related to the GNSS refractometry processing, such as the antenna height, base coordinate, antenna calibration parameters, reference ellipsoid, satellite availability and signal strengths. As GNSS refractometry is a relative observation method between the base and rover antenna, the processing is consistent for all data epochs and receivers, and there was no change in satellite signatures, we could exclude all such error sources. Since the offset is sudden and affects the receiver height, we assume a change in the effective physical height of the low-cost antenna. The antenna is screwed on the submerged lateral boom via a small vertical balise bar. Due to the overlying pressure of the snowpack and the cold, it could be possible that the screw has become loose and the antenna sank a few centimeters into the underlying snow. This can only be verified once the buried antennas are dug out in future.

The text is adapted to include this hypothesis.

---

## Author Comment (AC3)

We appreciate the very positive comment by David Shean, which shows the value of open review with public discussion opportunities. Regarding the comments made:

1. *There are a few additional relevant papers in the literature involving GNSS-Reflectometry for SMB of Antarctic ice shelves and ice streams:*

   *Shean et al. (2017) GPS-derived estimates of surface mass balance and ocean-induced basal melt for Pine Island Glacier ice shelf, Antarctica, https://tc.copernicus.org/articles/11/2655/2017/tc-11-2655-2017.html*

   *Seigfried et al (2017), Snow accumulation variability on a West Antarctic ice stream observed with GPS reflectometry, 2007–2017:*
   *https://agupubs.onlinelibrary.wiley.com/doi/full/10.1002/2017GL074039*

   *These papers offer some context, identify limitations, and provide further justification for the refractometry. I recommend you give them a read, and consider citing.*

   Thank you for these suggestions, we added them to the literature overview.

2. *I have not had a chance to do a detailed review, but I believe your experimental setup provides the (potentially unique) opportunity to characterize potential bias in snow measurements extracted from GNSS-Reflectometry alone. While I agree that GNSS-RR is a better option (assuming field support and equipment resources are available), there is also value in opportunistic reflectometry-only approaches leveraging existing archives of GNSS data collected by receivers deployed for other purposes (like ice motion). In other words, if you didn't have the refractometry, is there still value in the reflectometry results from your reference antenna alone?*

   The intention of our study was to extend a GNSS-IR setup with GNSS refractometry to study its potential for additional SWE and density estimation. However, before this study, the experimental site was only equipped with a reference antenna for GNSS-IR and tropospheric zenith path delay studies. There is definetily a value for the GNSS-IR results only and if you are interested in these data, they are publicly available and referenced in the paper. Additionally, ice movement, effective surface elevation and maybe ionospheric scintillation could be observed only by the reference antenna.

3. *I may have missed it, but it would be valuable to report the depth of the tower (or "sensor mast") in the firn at the start of the experiment. One important question is the depth of "bonding" between the tower and the firn, and whether the receivers rigidly mounted to the tower are experiencing relative downward motion due to compaction within upper or deeper layers of the firn column. Twit Conway has anecdotes about GNSS antenna poles near South Pole station penetrating several 10s of cm through plywood sheets due to differential firn compaction rates.*

   Thank you, this point was picked up in comment 2. of RC2, see our response there. Although such differential deformation might indeed be a problem for cold firn, the environmental conditions at Neumayer Station cause several melting events during summer to produce

considerable ice layering. The tower will thus be anchored in the firn by such refrozen melt layers.

4. *I believe your upward-looking "rover" antennas are rigidly attached to the same tower as the reference antenna, with an assumption that the rovers remain fixed relative to the original "firn surface" from the start of the experiment. If the tower (and all receivers) are moving downward at the same rate as the original firn surface, all is well, but if not (due to the tower bonding in deeper layers), then your "rover" antennas will be pulled below the layer corresponding to the original firn surface, and your derived measurements will also include upper layers of firn instead of just new snow accumulation. Hopefully that makes sense. If you can demonstrate that this is not an issue, that would be a nice addition!*

   This question was also picked up by both reviewers. Yes, the antennas are fixed with respect to each other and the tower and all measurements are thus relative.